# The Association between Maternal Sugar-Sweetened Beverage Consumption and Infant/Toddler Added Sugar Intakes

**DOI:** 10.3390/nu14204359

**Published:** 2022-10-18

**Authors:** Adrianne K. Griebel-Thompson, Abigail Murray, Katherine S. Morris, Rocco A. Paluch, Lisette Jacobson, Kai Ling Kong

**Affiliations:** 1Baby Health Behavior Lab, Division of Health Services and Outcomes Research, Children’s Mercy Research Institute, Children’s Mercy Hospital, 2401 Gillham Road, Kansas City, MO 64108, USA; 2Division of Behavioral Medicine, Department of Pediatrics, Jacobs School of Medicine and Biomedical Sciences, University at Buffalo, Buffalo, NY 14203, USA; 3Department of Population Health, University of Kansas School of Medicine-Wichita, Wichita, KS 67214, USA; 4Department of Pediatrics, University of Missouri-Kansas City, Kansas City, MO 64108, USA; 5Center for Children’s Healthy Lifestyles and Nutrition, University of Kansas Medical Center, Kansas City, KS 66103, USA

**Keywords:** infant, maternal, added sugars, sugar-sweetened beverage, obesity

## Abstract

Intake of added sugars during early life is associated with poor health outcomes. Maternal dietary intake influences the intake of their children, but little research investigates the relationship between maternal sugar sweetened beverage (SSB) and infant/toddler added sugar intakes. Our objective was to explore the relationship between maternal total sugars and SSB intakes and infant/toddler added sugar intakes. This cross-sectional study (*n* = 101) of mother-infant dyads measured maternal dietary intake by food frequency questionnaire and infant intake by three 24-h dietary recalls. Pearson’s correlations explored the relations between maternal total sugars and SSB intakes and infant added sugar intakes. Hierarchical stepwise regressions determined if maternal total sugars and SSB intakes explained the variation in infant added sugar intakes after accounting for known risk factors for early introduction of added sugars. Maternal total sugars (r = 0.202, *p* = 0.043) and SSB (r = 0.352, *p* < 0.001) intakes were positively correlated with infant/toddler added sugar intakes. In the hierarchical models, maternal total sugar intakes did not account for more variance in infant added sugar intakes (β = 0.046, *p* = 0.060), but maternal SSB intake was a significant contributor of infant added sugar intakes (β = 0.010, *p* = 0.006) after accounting for confounders. Interventions to reduced maternal SSB consumption may help reduce infant/toddler added sugar intakes.

## 1. Introduction

Due to the health risks associated with intake of added sugars in early life and childhood [1,2,3,4,5,6,7], the 2020–2025 Dietary Guidelines for Americans (DGA) recommend children ≤2 years of age avoid added sugars [8]. This guideline is in agreement with the 2017 Scientific Statement from the American Heart Association (AHA) which similarly recommends children under the age of 2 years avoid added sugars [7]. Added sugars are all sugars, syrups, and caloric sweeteners added during the food processing and manufacturing process [9]. Sources of added sugars include but are not limited to beverages, bakery foods, desserts, and candies [9]. Current research shows consumption of added sugars in children under 2 years of age is common. National Health and Nutrition Examination Survey (NHANES) data from 2011–2016, found 84% infants/toddlers (0–23 months) consume added sugars daily [2]. Early introduction of added sugars is also common with 22% of participants of the Special Supplemental Nutrition Program for Women, Infants, and Children (WIC) introduced to added sugars before 7 months of age [10]. Ha et al. reported a very similar proportion (1 out of 5) of infants had been introduced to added sugars by 6–9 months of age [11]. Only 13% of the WIC participants delayed introduction of added sugars until after 2 years of age [10].

Early introduction of added sugars may cause children to develop a preference for foods high in added sugars. The nutrients introduced early in life begin to shape future preferences for tastes, including sweetness [12]. The first source of nutrition for infants, human milk, is sweet [13], and to encourage consumption, a preference for sweetness is inherent [14,15]. This preference for sweetness increases with exposure as infants fed sugar water prefer more concentrated sugar solutions later in childhood [14,16,17]. Intake of added sugars increases with increasing age in the first 2 years of life which may further reinforce the preference for sweetness [18,19,20]. This is concerning as high intake of added sugars is associated with health problems. Early rapid weight gain [1], obesity [2,3], dental caries [2,4], asthma [2,5], cardiovascular disease [2,7], elevated blood pressure [2,6] and elevated triglycerides [2,6] are all known to be increased with high intake of added sugars during early life.

Identifying factors related to early added sugars consumption is important to find ways to reduce added sugar intakes. Income, number of children in the family, breastfeeding at 3 months, child sex, premature birth in addition to maternal age, educational attainment, employment, race/ethnicity, and weight status have been associated with early introduction of added sugars. [10,11]. One factor, maternal dietary intake, has received less attention. It is known that parental food habits determine child food choices [21], and that children’s intake reflects the intake of their mother [22]. A barrier to healthful eating in children is parental modeling of unhealthy behavior [23]. A few studies relate maternal dietary factors, intake of sweets during pregnancy or low fruit intake for example, to early introduction of added sugars [11,24], but whether maternal total sugars and SSB intakes influence this remains an understudied area.

The most compelling evidence that maternal SSB intake influences the added sugar intakes of their children comes from a cohort of Australian mother-infant dyads which found infants (6–9 months) of mothers who consumed SSB were 1.8 times more likely to have consumed foods and beverages with added sugars compared to those with mothers who did not consume SSB [11]. From this knowledge, our aim was to examine the association between maternal total sugars and specifically, SSB intakes and infant/toddler (age range 9–<16 months) added sugar intakes in a sample of mother-child dyads (*n* = 101). We hypothesized maternal total sugars and SSB intakes would be related to infant/toddler added sugar intakes.

## 2. Materials and Methods

### 2.1. Participants

This is a cross-sectional analysis of baseline data from 101 mother-infant/toddler dyads (infant/toddler aged 9 to <16 months) participating in a longitudinal randomized controlled trial [1]. Exclusion criteria included: infants born preterm (<37 weeks gestation), low-birth weight infants (<2500 g), infants/toddlers with known medical problems, infants/toddlers consuming special diets, infants/toddlers with developmental delays or disabilities, maternal smoking, alcohol abuse, or controlled substance use during pregnancy, mother <18 years of age, high risk pregnancy (Gestational Diabetes Mellitus, pre-eclampsia, etc.), or multiple gestation [1]. Three participants who consumed calories estimated to be ±2 SD from their estimated energy requirement (Institute of Medicine’s Food and Nutrition Board [25]) were not included in the present analysis. Trained study personnel collected informed consent from the parents of eligible participants, and participants were allowed to withdraw from the study at any time. The University at Buffalo Institutional Review Board approved this study.

### 2.2. Dietary Intake

#### 2.2.1. Maternal Dietary Intake

Maternal dietary intake was measured using the Block 2014 Food and Activity Questionnaire developed by NutritionQuest (Berkley, CA, USA) [26]. This is a full-length food frequency questionnaire (FFQ), which includes 127 foods and beverages, and specific questions to measure fat, carbohydrate, sugar, and whole grain intake [26]. The 2007–2008 and 2009–2010 NHANES cycles were used in the design of this FFQ [26] and the United States Department of Agriculture’s (USDA) Food and Nutrient Database for Dietary Studies (FNDDS 5.0), the Food Pyramid Equivalents Database (FPED), and the Nutrient Database for Standard Reference (SR27) were used to create a nutrient and food group analysis database for this FFQ [26]. There are two questions for each food or beverage [26]. The first question assesses how often the food or beverage is consumed [26]. The second question assesses the quantity of the food or beverage consumed at one time [26]. In this FFQ, SSB intake included all sugary beverages including fruit juice, soda, and energy drinks, among others [26].

#### 2.2.2. Infant/Toddler Dietary Intake

Infant/toddler dietary intake was measured using three caregiver-reported 24-h dietary recalls [1]. The 24-h dietary recalls were administered to parents of participants by phone on random occasions (two weekdays, one weekend day) within 10 days of anthropometric measurement, similar to the methodology used in the Feeding Infants and Toddlers Study (FITS) [27]. To ensure accurate reporting, parents were provided with a handout containing tips and answers to frequently asked questions to assist in reporting their child’s intake [1]. Prior to completing the recall, parents were asked if the infant/toddler had a normal/healthy eating day for the previous 24 h. If not, the recall would be completed on a new day [1]. All study personnel completed the dietary recalls using the United States Department of Agriculture’s Automated Multiple-Pass Method and were extensively trained by a registered dietitian (MS/RD) [28]. Additionally, all study personnel used a script when performing dietary recalls to ensure adherence to protocol [1]. Intake of supplements and medicines were not included in this analysis. See the previous publication by Kong et al. for details on recall methodology [1].

Parents were asked to keep a record of duration at the breast for breastfeeding participants to report intake of human milk. Nutritional intake from human milk was calculated by the FITS methodology [1,27]. Briefly, for exclusively breastfed infants (7–12 months), 600 mL of human milk was reported [1,27]. For those consuming both human milk and formula, the amount of formula reported was subtracted from 600 mL, and the remainder was reported as human milk [1,27]. For toddlers ≥ 12 months, one fl. oz. (29.6 mL) was reported for every 5 min at the breast [1,27].

Dietary intake data were collected and analyzed using Nutrition Data System for Research software version 2019, developed by the Nutrition Coordinating Center (NCC), University of Minnesota, Minneapolis, MN [29,30,31]. When study personnel could not find a specific food, a generic food found in the database was used. A total of 15 foods were not in the database and did not have a comparable generic food. In this case, the NCC was contacted, and the food was added to their database.

### 2.3. Demographic and Pregnancy History and Feeding Practices Questionnaires

Maternal and infant/toddler demographic information was collected including maternal age, sex, race and ethnicity, educational attainment, parity, and household size. As for infant/toddlers, we collected information on age, sex, and birth weight. We obtained pregnancy history and feeding practices information, such as breastfeeding duration and timing of introduction of solid foods, using the shortened version of the feeding questionnaire from the Infant Feeding Practices Study II [32]. This questionnaire consists of 13 questions which assess feeding practices including: initiation and duration of breastfeeding, timing of introduction of solid foods, infant/toddler age at cessation of breastfeeding, and pregnancy information of the mother [32]. The questionnaire was administered online using SurveyMonkey (https://www.surveymonkey.com, accessed on 24 May 2016).

### 2.4. Statistical Analysis

Pearson product-moment correlations were used to explore the association between (1) maternal total sugars and infant/toddler added sugar intakes and (2) maternal SSB intakes and infant/toddler added sugar intakes. Hierarchical stepwise regression models were used to test whether parent intake accounted for variability in infant/toddler added sugar intakes beyond other known predictors. In model 1, we controlled for child sex, child age (mo), birthweight (kg), parity, maternal body mass index (BMI) (kg/m^2^), gestational age, maternal education (y), and household income (USD 10K units) in step 1; we controlled for breastfeeding duration (mo) and first introduction of solid foods (mo) in step 2; and we controlled for maternal total sugar intakes in step 3. A similar hierarchical stepwise regression model was completed for model 2 with step 1 and step 2 being similar to the previous models and step 3 controlled for maternal SSB intake instead of maternal total sugar intakes. Incremental F-tests were used to determine if a statistically significant change occurred by comparing improvements in the model fit (R^2^). Statistical analysis was performed using SYSTAT 11 (Systat Software, Point Richmond, CA, USA, 2004) and SAS 9.4 (SAS Institute Inc., Cary, NC, USA, 2020).

## 3. Results

Demographic information for mothers and infant/toddlers can be viewed in Table 1. Briefly, this sample is predominately white (78%) and highly educated (72.3% had a bachelor’s degree or higher). These demographic factors may explain the high breastfeeding rate (68.3% breastfed for ≥6 m) and duration (8.1 ± 4.6 m) observed in this study. Average infant/toddler and maternal age were 11.9 ± 1.9 months and 32.6 ± 4.3 years, respectively.

Maternal total sugars (r = 0.202, *p* = 0.043) and SSB (r = 0.352, *p* < 0.001) intakes were positively correlated with infant/toddler added sugar intakes. Table 2 provides estimates from the stepwise hierarchical analysis. Step 1 accounted for 15.2% of the variance in infant/toddler added sugar intakes, and maternal BMI (β= 0.378, *p* = 0.014) and education (β= −1.265, *p* = 0.040) were significant predictors. The addition of breastfeeding duration (β = −1.356, *p* < 0.001) and first introduction to solids food (β = 0.695, *p* = 0.503) in step 2 increased R^2^ to 39.5%. While not significant, the final addition of maternal total sugar intakes in step 3 increased the R^2^ to 44.2% (β = 0.046, *p* = 0.060).

The second model (Table 3) has similar results to the first model for step 1 and step 2. In Step 3, maternal SSB intakes were added to the model, which explained 44.5% of the total variance in infant/toddler added sugar intakes. Maternal SSB intake was a significant contributor of infant/toddler added sugar intakes (*β* = 0.010, *p* = 0.006).

## 4. Discussion

This study sought to determine if maternal dietary intake, specifically total sugars and SSB intakes, are contributors to the added sugar intakes of their infants/toddlers. Some factors related to early introduction of added sugars are known while other factors, such as maternal dietary intakes, warrant further investigation. Early introduction of added sugars is more common among individuals with the following socioeconomic characteristics: low household income, low maternal educational attainment, shorter breastfeeding duration, and overweight and obesity (10, 11). While maternal total sugar intakes did not significantly contribute to infant/toddler added sugar intakes after accounting for known factors of early introduction of added sugars, maternal intake of SSB was a significant contributor of infant/toddler added sugar intakes. The FFQ used to collect maternal diet history in this study only provided data on total sugar intakes, which includes natural sources such as sugars from fruit and milk, not added sugar intakes. This may have explained the marginal significance we observed of maternal total sugars on infant/toddler added sugar intakes (β = 0.046, *p* = 0.060). Since we were not able to compare maternal added sugar intakes alone to infant/toddler added sugar intakes, we chose to compare a leading source of added sugars, SSB [34].

Similar research has been conducted by Ha et al. identifying increased odds of added sugar intakes in infants with mothers who consume SSB [11]. While the objective of the study by Ha et al. is relatively similar to ours, the diet history collection methods were different. Ha et al. measured dietary and SSB intake by questionnaires for both infants and mothers [11]. In the current study, we used the preferred methodology, three 24-h dietary recalls [35], to collect dietary intake in infants/toddlers and a FFQ to collect maternal dietary intake. The use of three 24-h dietary recalls to collect dietary intakes provide an improved estimation of intake of added sugars. Due to this difference, we were able to measure added sugars as a continuous variable compared to the binary variable of early introduction of added sugars in the study by Ha et al. [11]. Though different, these studies suggest women who consume beverages with higher amounts of sugars also feed their children higher amounts of added sugars. This provides evidence that negative modeling of SSB consumption by mothers leads to increased added sugar intakes among their children. As preferences for food develop early in life, and infants have a predisposition for sweetness [14,15], delaying the introduction of added sugars until after the second year of life may work to counteract this preference of sweetness. The DGA [8] and AHA [7] recommend the avoidance of added sugars in the first 2 years of life due to the health risks associated with early introduction of added sugars [1,2,3,4,5,6,7].

Interventions have been conducted to reduce SSB consumption. At the societal level, taxation of SSB [36], warning labels included to the front of SSB [37], and school-based interventions [38,39] have been incorporated in many countries. At the individual level, interventions that provide or encourage water consumption, such as providing water filters to clean tap water [40], or education programs on SSBs and healthy lifestyle choices [41,42] have reduced SSB consumption. Other studies have used behavior change theories to decrease SSB consumption [43,44,45,46]. For example, the Smart Moms intervention [43,44] used social cognitive theory, a type of behavior change theory, to design a phone-based program to decrease SSB and juice intake in mother-child (3–5 years of age) dyads. The Smart Moms intervention significantly decreased maternal caloric beverage intake, increased limiting setting for 100% fruit juice for their children, and decreased SSB/fruit juice consumption of the children [43,44]. Thus, interventions targeting maternal SSB consumption may decrease the sugar intakes of their children. Contrary to these studies, other behavior interventions targeted at mothers did not reach statistical significance in decreasing child SSB intakes [45,46]. For example, there was no difference in child SSB intake in a group of children whose mothers received maternal emotional regulation, home environment, feeding practices, and healthy behavior modeling interventions compared to a control group [45]. The study by Taveras and colleagues, which provided motivational interviewing and incentive kits to families, found no effect on SSB intake [46].

Against this background, perhaps more innovative and targeted interventions to reduce the SSB intake of mothers are needed. It is possible that an appropriate setting for maternal SSB consumption interventions may be during well-child check-ups at pediatrician offices, or primary-care appointments for the mothers. Physicians, nurses, or dietitians in these settings could provide education on SSB and counseling on strategies to decrease maternal SSB consumption. WIC may also be an appropriate setting for this type of intervention. In 2018, 45% of infants born in the United States and 61% of 1-year old children were enrolled in the WIC program [47]. Due to the wide coverage of women and children and the previously shown benefits of WIC participation on infant/toddler added sugar intakes [48], the WIC office may be an ideal setting for maternal SSB interventions. Another unique factor of WIC participation is that both the mother and the infant/toddler are clients of WIC as opposed to physician appointments which are for either mother or infant/toddler. Furthermore, the WIC nutritionist might have already built rapport with the mother during pregnancy appointments, increasing the trust between the client and the WIC nutritionist. Finally, WIC participation has already been shown to have a beneficial impact on added sugar intakes of infants/toddlers [48], so the addition of interventions to decrease SSB consumption targeted towards mothers during WIC visits may further decrease infant/toddler added sugar intakes.

Our research supports the hypothesis that maternal dietary intake influences the diet of their children [49,50], and expands on the literature by focusing on (1) the influence of maternal SSB consumption on infant/toddler added sugar intakes and (2) a sample of children under the age of 2 years. The dietary recall methodology used to collect infant/toddler diet history is a strength of this study. Collecting a total of three 24-h dietary recalls, as done in this study, is the preferred standard [35]. This paired with the Automated Multiple-Pass Method and extensive training by an MS/RD for collecting the data increases the confidence that we collected accurate diet histories. A limitation is the maternal report of infant/toddler intake in which overestimation is likely [51]. As for demographics, we have a predominately white, and high SES cohort, and results may not be generalizable.

## 5. Conclusions

The current study begins to address the gap in knowledge on how specific maternal dietary intake, such as SSB, influences the dietary intakes of their offspring. Our data suggests maternal SSB consumption is a significant contributor to infant/toddler added sugar intakes above and beyond other known factors (i.e., demographics and maternal BMI). Research is needed to better understand the preference for and high consumption of sweet food and drinks in some individuals but not others. Our study is the first step in examining the environments of infants/toddlers in relation to their added sugar intakes. This study explored infant/toddler added sugar intakes in relation to maternal behavior/modeling, but future studies need to address how early exposure to added sugars through feeding practices, i.e., feeding of infant formula, relates to added sugar intakes later in childhood. Additionally, findings from observational studies such as this could be used to inform intervention development. For example, in order to limit infant/toddler added sugar intakes, designing an intervention to reduce maternal SSB consumption may be indicated.

## Figures and Tables

**Table 1 nutrients-14-04359-t001:** Participant characteristics (*n* = 101).

	Mean (SD)	N (%)	Range
**Child**			
Sex, male		45 (44.6)	
Age, mo	11.9 ± 1.9		9.1–15.8
Race, white		87 (78.0)	
Refuse to answer		0(0.0)	
Gestational age, weeks	39.4 (1.2)		37–42
Birth weight, kg	3.5 ± 0.5		2.4–5.2
Weight-for-length z-score ^a^	0.5 ± 0.9		−1.7–3.1
Weight-for-age z-score ^a^	0.2 ± 0.9		−2.4–2.6
Length-for-age z-score ^a^	−0.3 ± 1.2		−3.1–2.9
Conditional weight gain ᵇ	0 ± 1.0		−2.8–2.3
Breastfeeding duration	8.1 (4.6)		0–12.0
≥6 mo		69 (68.3)	
First introduction to solid foods	5.3 ± 1.0		2.0–9.0
<4 mo		3 (3.0)	
4–5 mo		42 (41.6)	
≥6 mo		74 (55.4)	
**Mother**			
Age, y	32.6 ± 4.3		22.8–46.3
Education level			
Some college or below		28 (27.7)	
College graduate or higher		73 (72.3)	
Refuse to answer		0 (0.0)	
Parity			
Nulliparous		56 (55.4)	
Parous ≥ 1		45 (44.6)	
Current BMI, kg/m^2^	30.2 ± 7.7		19.6–49.3
Normal weight		33 (33.3)	
Overweight/obese (≥25 BMI)		68 (67.30)	
Household total income		USD 91,386 (41,328)	
<USD 30,000		6 (5.9)	
USD 30,000–USD 69,999		24 (23.8)	
USD 70,000–USD 109,999		43 (42.6)	
≥USD 110,000		28 (27.7)	
Refuse to answer		0 (0.0)	

SD = Standard Deviation; ^a^ Calculated by using the WHO growth charts; ᵇ Calculated using method described by Griffiths et al., 2009 [33].

**Table 2 nutrients-14-04359-t002:** Hierarchical regression model of maternal total sugar intakes (*R*^2^, change statistics [Δ*R*^2^], and regression coefficients [*β*]) predicting infant added sugar intakes (*n* = 101).

Effect	*R* ^2^	Δ*R*^2^	*β*	t	*p*-Value
**Step 1**					
Child sex			0.731	0.313	0.755
Child age (mo)			−0.119	−0.192	0.848
Birthweight (kg)			−4.418	−1.754	0.083
Parity			0.248	0.227	0.821
Maternal BMI (kg/m^2^)			0.378	2.508	0.014 *
Gestational age			0.004	0.004	0.997
Maternal Education (y)			−1.265	−2.082	0.040 *
Household income (USD 10K)			0.422	1.473	0.144
	0.152				
**Step 2**					
Breastfeeding duration (mo)			−1.356	−5.898	<0.001 *
First introduction to solid foods (mo)			0.695	0.672	0.503
Finc (2,90) =18.07, *p* < 0.0001	0.395	0.240			
**Step 3**					
Maternal total sugar intakes	0.442	0.030	0.046	1.904	0.060
Finc (1,89) =7.49, *p* = 0.007					

* Indicates *p* < 0.05.

**Table 3 nutrients-14-04359-t003:** Hierarchical regression model of maternal sugar sweetened beverage intakes (*R*^2^, change statistics [Δ*R*^2^], and regression coefficients [*β*]) predicting infant added sugar intakes (*n* = 101).

Effect	*R* ^2^	Δ*R*^2^	*β*	t	*p*-Value
**Step 1**					
Child sex			0.731	0.313	0.755
Child age (mo)			−0.119	−0.192	0.848
Birthweight (kg)			−4.418	−1.754	0.083
Parity			0.248	0.227	0.821
Maternal BMI (kg/m^2^)			0.378	2.508	0.014 *
Gestational age			0.004	0.004	0.997
Maternal Education (y)			−1.265	−2.082	0.040 *
Household income (USD 10K)			0.422	1.473	0.144
	0.151				
**Step 2**					
Breastfeeding duration (mo)			−1.356	−5.898	<0.001 *
First introduction to solid foods (mo)			0.695	0.672	0.503
Finc (2,90) =18.07, *p* < 0.0001	0.395	0.240			
**Step 3**					
Maternal sugar sweetened beverage intakes	0.445	0.054	0.010	2.831	0.006 *
Finc (1,89) = 8.02, *p* = 0.006					

* Indicates *p* < 0.05.

## Data Availability

Data can be provided upon request.

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
