# Peer review of "The Association between Maternal Sugar-Sweetened Beverage Consumption and Infant/Toddler Added Sugar Intakes"

_nutrients, 2022, doi:10.3390/nu14204359_

Round 1

Reviewer 1 Report

This is an original article that investigated the association between maternal SSB consumption and infant/toddler added sugar intakes. The authors presented that maternal SSB consumption significantly contributes to infant/toddler added sugar intakes. The topic of the maternal diet is wide interest in the health risks of children, and only a few studies have been reported on the maternal SSB intake.

However, the number of participants was too small to support conclusions with this study. For example, the number of participants in cited paper number 2, 3, 4, and 5, are 1,211, 1,165, 16,508, 15,960, respectively. This study has only low statistical power, and should be re-analyzed with a larger number of participants.

Author Response

Reviewer 1:

This is an original article that investigated the association between maternal SSB consumption and infant/toddler added sugar intakes. The authors presented that maternal SSB consumption significantly contributes to infant/toddler added sugar intakes. The topic of the maternal diet is wide interest in the health risks of children, and only a few studies have been reported on the maternal SSB intake.

We thank the reviewer for the positive comment.

However, the number of participants was too small to support conclusions with this study. For example, the number of participants in cited paper number 2, 3, 4, and 5, are 1,211, 1,165, 16,508, 15,960, respectively. This study has only low statistical power, and should be re-analyzed with a larger number of participants.

We appreciate this feedback. Our sample size is smaller than the sample size in references 2, 3, 4, and 5, but we used the preferred methodology of three 24-hr dietary recalls to collect dietary intake in a population of infants/toddlers ages 9-<16 m. We also used the United States Department of Agriculture's Automated Multiple-Pass Method to obtain the 24-hr dietary recalls and study personnel were extensively trained by a graduate-level trained registered dietitian. Finally, we used the Nutrition Data System for Research (NDSR) to analyze nutrient intakes and specifically, added sugar intakes. Due to this, the data we collected is of high quality, and we do not need a large sample size to detect significance. In comparison, reference 2 collected dietary intake by only 1 day 24-hr dietary recall in infants/toddlers 0-23 m. Reference 3 collected dietary intake by only 2 day 24-hr dietary recalls in children 2-18 years. References 4 and 5 collected dietary intake by questionnaire in children 5-16 years and students from 9-12th grade, respectively. We agree with the reviewer that a lack of observed significance between maternal total sugar intakes and infant/toddler added sugar intakes could be due to our sample size. However, our effect size of f2=0.09 for the relationship between maternal SSB consumption and infant/toddler added sugar intakes had observed power of 0.84. In addition, our effects for breastfeeding and introduction of solid foods were large (f2=0.40) and had power of 0.99.

Reviewer 2 Report

The article's topic is very interesting, and its strength lies in its topicality. My recommendations are the following:

I'm concerned about calculations based on breastfeeding duration. As I can see SD for this parameter is 4.6 while the mean is 8.1.

The conclusion section could be more developed. It would be great to show your future research plans based on this research.

Author Response

Reviewer 2:

The article's topic is very interesting, and its strength lies in its topicality. My recommendations are the following:

We thank the reviewer and agree that this topic is timely.

I'm concerned about calculations based on breastfeeding duration. As I can see SD for this parameter is 4.6 while the mean is 8.1.

Thank you for this comment. We have looked into the breastfeeding duration variable, and the reported data are correct. It is possible the standard deviation is wide because the duration of breastfeeding varies from no breastfeeding at all to currently breastfeeding at time of contact. In addition, this is a predominately non-Hispanic white sample with high income and educational attainment, all of which are demographic factors related to higher breastfeeding rates. We believe the demographics of our sample explain the high breastfeeding rate and duration reported in this study. The results section has been adjusted to explain this (lines 172-173).

The conclusion section could be more developed. It would be great to show your future research plans based on this research.

The conclusion has been expanded to include our future plans on this research topic (line 281-286).

Reviewer 3 Report

Comments to the Authors of manuscript number: nutrients-1946034 entitled “The association between maternal sugar-sweetened beverage consumption and infant/toddler added sugar intakes”.

The paper is very well written and the discussion about presented problem is needed, especially when from year to year the number of obese children rises. Many various factors have to be analyzed as it is presented by Authors who in consistent manner have presented the association between the behavior of mother and children in the case of the preference of sweet taste.

It is very rare situation when I do not have comments and in the peer review process I recommend to accept in the first step.

1. The introduction presents the current problem with added sugars in children below 2 years and the way by which various health organizations want to prevent before the health risks associated with added sugar in early life.

2. Next, Authors explained why children prefer sweet taste and health problems in these children which can occur later in life.

3. Many factors relating to added sugar are presented

4. Maternal and children dietary intake is very well described in many details.

5. Authors have also collected numerous demographic information, important to understand the problem

6. Proper statistical analysis was performed.

7. The discussion analyzes the problem in a logical manner in comparison to other previously conducted studies.

Author Response

Reviewer 3:

Comments to the Authors of manuscript number: nutrients-1946034 entitled “The association between maternal sugar-sweetened beverage consumption and infant/toddler added sugar intakes”.

The paper is very well written and the discussion about presented problem is needed, especially when from year to year the number of obese children rises. Many various factors have to be analyzed as it is presented by Authors who in consistent manner have presented the association between the behavior of mother and children in the case of the preference of sweet taste.

It is very rare situation when I do not have comments and in the peer review process I recommend to accept in the first step.

  1. The introduction presents the current problem with added sugars in children below 2 years and the way by which various health organizations want to prevent before the health risks associated with added sugar in early life.
  2. Next, Authors explained why children prefer sweet taste and health problems in these children which can occur later in life.
  3. Many factors relating to added sugar are presented
  4. Maternal and children dietary intake is very well described in many details.
  5. Authors have also collected numerous demographic information, important to understand the problem
  6. Proper statistical analysis was performed.
  7. The discussion analyzes the problem in a logical manner in comparison to other previously conducted studies.

We thank the reviewer for the positive feedback.

Round 2

Reviewer 1 Report

The result and conclusion had been improved with more explanations.